# Cerivastatin Synergizes with Trametinib and Enhances Its Efficacy in the Therapy of Uveal Melanoma

**DOI:** 10.3390/cancers15030886

**Published:** 2023-01-31

**Authors:** Adriana Agnese Amaro, Rosaria Gangemi, Laura Emionite, Patrizio Castagnola, Gilberto Filaci, Martine J. Jager, Enrica Teresa Tanda, Francesco Spagnolo, Matteo Mascherini, Ulrich Pfeffer, Michela Croce

**Affiliations:** 1IRCCS Ospedale Policlinico San Martino, 16132 Genova, Italy; 2Centre of Excellence for Biomedical Research, Department of Internal Medicine, University of Genoa, 16132 Genova, Italy; 3Leiden University Medical Center, 2333 ZA Leiden, The Netherlands

**Keywords:** uveal melanoma, MEK inhibitor, statins, YAP/TAZ

## Abstract

**Simple Summary:**

Uveal melanoma is a rare and aggressive disease. Gα-proteins GNAQ and GNA11 are driver mutations that activate MAP kinase and YAP/TAZ pathways. *BAP1* loss and monosomy of chromosome 3 are present in patients with high risk of metastasis. MEK-inhibitors do not significantly block UM progression. Combinations of the MEK inhibitor trametinib and different classes of drugs targeting YAP/TAZ were used to overcome resistance. Combination of trametinib and cerivastatin were synergistic in vitro and in vivo in *BAP1* mutated and chromosome 3 monosomic uveal melanoma cell lines.

**Abstract:**

Background: Metastatic uveal melanoma (MUM) is a highly aggressive, therapy-resistant disease. Driver mutations in Gα-proteins GNAQ and GNA11 activate MAP-kinase and YAP/TAZ pathways of oncogenic signalling. MAP-kinase and MEK-inhibitors do not significantly block MUM progression, likely due to persisting YAP/TAZ signalling. Statins inhibit YAP/TAZ activation by blocking the mevalonate pathway, geranyl-geranylation, and subcellular localisation of the Rho-GTPase. We investigated drugs that affect the YAP/TAZ pathway, valproic acid, verteporfin and statins, in combination with MEK-inhibitor trametinib. Methods: We established IC50 values of the individual drugs and monitored the effects of their combinations in terms of proliferation. We selected trametinib and cerivastatin for evaluation of cell cycle and apoptosis. Synergism was detected using isobologram and Chou–Talalay analyses. The most synergistic combination was tested in vivo. Results: Synergistic concentrations of trametinib and cerivastatin induced a massive arrest of proliferation and cell cycle and enhanced apoptosis, particularly in the monosomic, *BAP1*-mutated UPMM3 cell line. The combined treatment reduced ERK and AKT phosphorylation, increased the inactive, cytoplasmatic form of YAP and significantly impaired the growth of UM cells with monosomy of chromosome 3 in NSG mice. Conclusion: Statins can potentiate the efficacy of MEK inhibitors in the therapy of UM.

## 1. Introduction

Uveal melanoma (UM), the most common primary intraocular tumour in adults, is characterized by marked variability in its ability to metastasize. Up to half of the patients with UM develop distant metastases, most commonly in the liver (for a recent review, see [1]). Unlike cutaneous melanoma, the mutational burden in UM is low, with a mean of only 17 [2] to 30 [3] non-synonymous mutations in protein-coding sequences per exome or 0.5 mutations per megabase [4]. UM is molecularly distinct from cutaneous melanoma (CM), and the two diseases are characterized by entirely different driver mutations; *B-Raf* Proto-Oncogene (*BRAF*), *NRAS* Proto-Oncogene (*NRAS*), and *Neurofibromin 1* (*NF1*) [5] in CM; and *G Protein Subunit Alpha Q* (*GNAQ*) and *11* (*GNA11*), and *BRCA1- associated Protein 1* (*BAP1*) [6,7,8] in UM. Co-driver gene mutations in the *Protein Tyrosine Kinase 2 Beta* (*PTK2B*) have been identified [9]. Mutations in the *alpha G-protein subunits GNAQ* and *GNA11*, are found in most primary UM in a mutually exclusive pattern [10]. Mutations in *BAP1*, *Splicing Factor 3b Subunit 1* (*SF3B1*), and *Eukaryotic Translation Initiation Factor 1A X-Linked* (*EIF1AX*) are associated with high, intermediate, and low metastatic risk, respectively [11]. BAP1, a member of the ubiquitin C-terminal hydrolase subfamily of deubiquitinating enzymes, deubiquitinates mono-ubiquitinated histone H2A at K119 in *Polycomb* gene repression [11,12,13], acting as a tumour suppressor. Both alleles are inactivated in high-risk UM, commonly by loss of one copy of chromosome 3 and mutation of the remaining *BAP1* allele.

GNAQ and GNA11 activate the classical G-protein signalling cascade via inositol-3-phosphate, diacyl-glycerol, and cyclic AMP leading to the stimulation of mitogen-activated protein (MAP) kinases, protein kinase B (Akt), and C (PKC), phosphoinositide 3-kinase (PI3K) and the mechanistic target of rapamycin (mTOR) [14]. Despite evidence of the functional importance of the MAPK pathway, MAPK-targeted therapy has been unsuccessful in UM [15], suggesting the involvement of other oncogenic pathways. GNAQ and GNA11 have indeed been shown to activate the transcription factor complex YAP/TAZ in a HIPPO-independent manner [16,17]. The HIPPO-YAP/TAZ pathway has been identified as an important regulator of organ size, and its involvement in several cancer types has recently been described [18] and is associated with resistance to chemotherapy [19] and BRAF/MEK/EGFR-targeted therapies [20,21].

YAP/TAZ activity is under the direct control of cell shape and polarity dictated by the cytoskeletal structure [22]. YAP/TAZ proteins in normal tissues are retained in the cytoplasm through phosphorylation at specific serine residues [23]. However, in cancer, YAP/TAZ proteins are translocated into the nucleus, where they bind to the TEA-domain transcription factor (TEAD), which induces transcription of proliferative genes and inhibition of pro-apoptotic genes [23].

Statins are potent inhibitors of the mevalonate pathway and are used in clinical practice to prevent hypercholesterolemia-derived cardiovascular and coronary heart diseases [24]. The mevalonate is an important metabolic pathway that produces sterols and isoprenoids, such as farnesyl pyrophosphate (FPP) and geranylgeranyl pyrophosphate (GGPP) involved in the prenylation of small GTPases. GGPP activates YAP by inhibiting its phosphorylation and inducing its nuclear translocation. Statins inhibit the rate-limiting enzyme of the mevalonate pathway, 3-hydroxy-3-methylglutaryl-CoA reductase (HMGCR), and play an anti-cancer role through the reduction of isoprenoids [25].

Statins, used to inhibit the HMGCR activity, also antagonise isoprenylation and inhibit nuclear localisation and transcriptional activity of YAP/TAZ, mediating a potential anti-cancer activity [26,27]. A possible role in the prevention of melanoma metastasis has been proposed for statins through changes in the transcriptome of human melanoma cell lines [28].

The main oncogenic functions of YAP/TAZ have been attributed to their interaction with the TEAD transcription factor. The ophthalmological drug verteporfin, an inhibitor of the YAP/TEAD interaction, has shown efficacy in vitro on UM cell lines, except on those bearing *BAP1* mutation [29].

Histone acetylation controls gene expression by modulating the access of transcription factor components to genomic regions. The histone deacetylases (HDACs) levels are increased in cancer [30]. Target genes regulated by HDACs are involved in proliferation, apoptosis, and immunogenicity. Their overexpression in cancer indicates them as therapeutic targets [31]. Several HDAC inhibitors (HDACi) have been developed. One of these, valproic acid, affects the activity of two of the four HDAC classes, namely HDAC-I and -II, and stimulates apoptotic cell death in human melanoma cells [30]. HDACi exert inhibitory effects in UM cells, including *BAP1*-mutated cells [32,33]. A phase II trial of the HDACi vorinostat in patients with advanced UM has recently completed accrual (NCT01587352).

With the aim to identify drug combinations that can be used in the treatment of metastatic UM, we analysed those expected to inhibit both MAP-kinase and YAP-signalling and to block UM proliferation. In particular, we focused on the effects of statins and used verteporfin and valproic acid drugs that potentially block YAP-signalling in parallel for comparison. Verteporfin is used to treat macular degeneration and valproic acid is under clinical investigation for adjuvant therapy of UM. The most potent statin and trametinib combination, cerivastatin and trametinib, was tested in vivo using UPMM3, a cell line with monosomy of chromosome 3 (Chr3) and a loss-of-function mutation of the remaining *BAP1* allele, as a model.

## 2. Materials and Method

### 2.1. Cancer Cell Lines

The human UM cell lines 92.1 [34], MEL270 [35], OMM1 [36], OMM2.5 [37], UPMM3 [38], and UPMM2 [38] were cultured in Roswell Park Memorial Institute 1640 medium (RPMI, Life Technologies Corporation, San Francisco, CA, USA), supplemented with 10% foetal bovine serum, 100 U/mL Penicillin-Streptomycin (Life Technologies Corporation, San Francisco, CA, USA), 2 mM L-glutamine (Life Technologies Corporation, San Francisco, CA, USA). Cells were cultured in a humidified incubator at 37  °C and supplemented with 5% CO_2_.

### 2.2. Reagents

Trametinib (Selleckchem, Planegg, Germany) was dissolved in DMSO at a final concentration of 40 mM, aliquoted, and kept at −80 °C. Atorvastatin, simvastatin, cerivastatin, and verteporfin (MedChemExpress, Sollentuna, Sweden) powders were dissolved with DMSO at a final concentration of 10 mM, aliquoted, and kept at −20 °C. Valproic acid (Sigma-Aldrich, Milan, Italy) was dissolved in deionised water at a final concentration of 1 M and kept at −20 °C.

### 2.3. Drug Screening Assay

Cells were seeded in 96-well, flat-bottom plates (Corning, Merck Life Science srl, Milan, Italy). The following day, increasing drug concentrations and their combinations were added to the cell culture and incubated for 72 h. Cell viability was tested using a 3-(4,5-dimethylthiazol-2-yl)-2, 5-diphenyl tetrazolium bromide (MTT) assay, (Sigma-Aldrich, Milan, Italy). Optical density (OD) was read on a multiwell scanning spectrophotometer (ELISA reader BioTek, Savatec, Torino, Italy) at 570 nm. All experiments were performed three times. The inhibitory concentration that led to 50% growth reduction (IC50) was calculated using a wide range of drug concentrations by the Graph Pad Prism v 9.4 software (San Diego, CA, USA). The synergism evaluation was performed using the MTT cell viability results obtained after single and combined treatments. The MTT assay was used to measure cellular metabolic activity as an indicator of cell viability, proliferation and cytotoxicity. Since none of the drugs were expected to directly affect mitochondrial activity, the formation of formazan can be used as a proxy for proliferation in UM cell lines. Drug combinations were evaluated following the method established by Chou and Talalay [39] and the concept of combination index (CI) CI = D1/E1 + D2/E2. D1 and D2 are the actual IC50 drug doses in the combinations during dosing experiments and E1 and E2 are individual IC50 drug levels. CI gives a quantitative definition of synergism (CI < 1), additive effect (CI = 1), and antagonism (CI > 1).

### 2.4. Cell Cycle Analysis

UM cells were seeded in 24-well plates and treated with either trametinib and cerivastatin at IC50 or different synergistic concentrations alone or in combination for 72 hrs. Cells were harvested and 1 × 10^5^ cells were washed in PBS, permeabilised in cold 70% ethanol, and incubated overnight at 4 °C in the dark. After washing with PBS, the cells were incubated in 1 ml of propidium iodide (PI) staining solution for 30 min at RT in the dark. The cell cycle was determined using a flow cytometer (FACScan; Becton & Dickinson Italy, Milan, Italy) and analysed by ModFit LT v3.0 (Beckman Coulter, Indianapolis, IN, USA) software.

### 2.5. Annexin V-FITC/PI Apoptosis Analysis

Apoptosis was detected with Annexin V-fluorescein isothiocyanate (FITC) and PI, accordingly to the manufacturer’s instructions (eBioscience, Thermo Fisher Scientific, Waltham, MA, USA), and analysed by FACScan Flow cytometer and CellQuest (BD Biosciences, San Jose, CA, USA) software.

### 2.6. Western Blotting Analysis

Total proteins were extracted using Complete Lysis buffer containing, proteases inhibitors (04719956001 Roche) and phosphatase inhibitors (04906845001 Roche). Preparation of nuclear extracts was performed by using NE-PER™ nuclear and cytoplasmic extraction reagents according to the manufacturer’s instructions. The Western blot of cell lysates was performed as previously described [40]. Antibodies: p-YAP rabbit mAb (#13008), YAP rabbit polyclonal (#4912), P-FAK rabbit (#2383), Cleaved caspase-3 rabbit mAb (#9664), cleaved PARP rabbit polyclonal (#9541), and P-AKT rabbit mAb (#4060) all purchased by Cell Signalling (MA, USA); and P-ERK mouse mAb (#sc-7383, Santa Cruz Biotechnology, USA), HDAC1 rabbit Ab (#H3284, Sigma-Aldrich), anti-β-tubulin (HRP) rabbit polyclonal (ab21058) and anti-GAPDH rabbit polyclonal antibodies (ab9385) from Abcam (Cambridge, MA, USA). Antibody binding was revealed by ECL Prime (RPN2232, GE Healthcare, Milan, Italy) and a chemiluminescence gel documentation and analysis system (MINI HD, UVITEC, Cambridge, UK).

### 2.7. Xenograft Mouse Model

Six-weeks old NOD.Cg-Prkdc^scid^ Il2rg^tm1wjl^/SzJ mice were obtained from Charles River (Charles River Laboratories Italia Srl). The animals were housed in pathogen-free conditions, and experiments were performed according to the National Regulation on Animal Research Resources and approved by the Institutional Review Board (Aut n°190/2021-PR (risp prot.22418.151)). UPMM3 cells (10^7^) were injected subcutaneously in 16 mice that were randomised in 4 groups of 4 mice each. Trametinib and cerivastatin were initially dissolved in DMSO and diluted into an aqueous pooled dose containing a final concentration of 0.5% hypromellose (Sigma-Aldrich) and 0.05% Tween-80 (Sigma), in saline. Control animals received the vehicle. Final maximal DMSO concentration was 8.8% *v*/*v*. Mice weight and tumours were measured weekly. Tumour volume was calculated as follows: V = ½ × L × W × H.

### 2.8. Microarray

UPMM3 cells were treated for 24 h with trametinib 10 nM, or cerivastatin 0.125 µM or the combination of these drugs. The experiment was repeated three times. RNA was extracted from cell lines using RNeasy Plus mini kit (Qiagen, Hilden, Germany). RNA quality was assessed with Nanodrop and BioAnalyser tools (Agilent, St. Clara, CA, USA). cDNA, ds-cDNA, and cRNA synthesis and fragmentation were performed using the 3’ IVT Express Kit (Affymetrix, Santa Clara, CA, USA). Hybridisation, washing, and staining were performed using the GeneAtlas (Affymetrix, St. Clara, CA, USA). All micro-array data are MIAME compliant. The dataset, corresponding to 12 cell files is available from the GEO database (http://www.ncbi.nlm.nih.gov/geo/, accessed on 26 January 2023), under accession number GSE212219.

### 2.9. Statistical Analysis

The paired or unpaired Student’s *t*-test was used when appropriate. *p* values are shown as following: (* *p* < 0.05; ** *p* < 0.01; *** *p* < 0.001). Analyses were performed using PRISM v 9.4 (Graph pad, La Jolla, CA, USA).

Micro-array gene expression data were analysed in R/BioConductor. Quantile normalisation was performed using RMA [41], differentially expressed genes were identified by significance analysis of micro-arrays [42], applying variance and intensity filters. Significant genes (FDR = 0) were clustered by applying hierarchical clustering with average linkage and Euclidean distance measure as described earlier [43].

## 3. Results

### 3.1. Anti-Proliferative Activity of MEK and YAP/TAZ Inhibitors in UM Cell Lines

MAPK-targeted therapy has so far not been successful in UM as a single agent. With the objective to find drugs that can be combined with inhibitors of MAPK to improve their effectiveness, we decided to evaluate the cytotoxic effects of drugs targeting the oncoprotein YAP/TAZ in UM cell lines because the YAP/TAZ pathway plays an important role in UM development (see also in the discussion section). We assessed the effects of statins that target the mevalonate/cholesterol biosynthetic pathway and interfere with YAP protein activation and that inhibit YAP nuclear translocation [27]. We also addressed the effects of verteporfin, a benzoporphyrin derivative, that interferes with the interaction of activated YAP with TEAD within the nucleus [44] and valproic acid (VPA), which belongs to the class of HDACi known to also inhibit YAP and AKT signalling [45]. Valproic acid and verteporfin have already been tested in UM cell lines. Recently, Faião-Flores et al. demonstrated that HDACi show synergistic anti-metastatic effects if associated with MEK inhibitors in UM [45]. Brower et al. described verteporfin cytotoxic activity in vitro on most UM cell lines [29].

We selected the following 6 human *bona fide* UM cell lines with *GNAQ* or *GNA11* mutations (Table 1): 92.1, MEL270, UPMM2, UPMM3, OMM1, and OMM2.5, bearing different genetic characteristics, derived from either primary (MEL270, 92.1, UPMM2 and UPMM3) UM or metastatic (OMM1 and OMM2.5) UM. The cell line 92.1 shows three chromosomes 3 and wt *BAP1*. MEL270, OMM1, and OMM2.5 are disomic for chromosome 3 and wt for *BAP1*, and UPMM2 and UPMM3 are monosomic for chromosome 3 and *BAP1*-mutated. The cell lines MEL270 (primary UM) and OMM2.5 (MUM) were derived from the same patient.

Cells were treated with different concentrations of each drug for 72 h. Trametinib showed an IC50 value in the nanomolar range of concentrations. The cell line MEL270 was the most sensitive cell line, with the lowest IC50 (5 nM ± 1.4, mean ± SEM), OMM1 and OMM2.5 were the least sensitive cell lines (6500 nM ± 3983 and 171.6 nM ± 44, respectively) with OMM1 showing the highest IC50 (Table 2). Primary disomic cells were more sensitive to trametinib than monosomic cells, confirming previous in vitro results [46], whereas metastatic cell lines were highly resistant. VPA and HDACi have been widely studied in UM. In our study, VPA showed to be effective against UM cell lines only at concentrations in the millimolar range, with IC50 ranging from 3.39 mM ± 0.98 in 92.1 to 40.9 mM ± 26.5 in OMM2.5 (Table 2). Verteporfin showed activity on UM cell lines at micromolar concentrations, with UPMM2 being the least sensitive of all (720.6 µM ± 703.5) (Table 2). Among statins, cerivastatin decreased proliferation of UM cell lines at very low concentrations, ranging from 0.06 to 1.56 µM (Table 2). Simvastatin and atorvastatin showed a higher IC50 compared with cerivastatin. Therefore, cerivastatin was selected for drug combination experiments.

### 3.2. In Vitro Anti-Proliferative Activity of Trametinib-Based Combinations, in UM Cell Lines

We tested the combination of trametinib with cerivastatin combining both drugs at concentrations below IC50. We used the method described by Chou and Talalay [39] to calculate the combination index (CI) for additive effect (CI = 1), synergism (CI < 1), and antagonism (CI > 1). The experiments were performed on those cell lines that showed a detectable IC50 for the single drugs. Trametinb and cerivastatin showed synergistic activity in UPMM2 and UPMM3 cells, 72 h after the beginning of treatment. The combination of trametinib and cerivastatin showed synergistic activity in OMM2.5 cell line after one week of treatment (Figure 1 and Table 3). In addition, we verified the possible synergistic activity of trametinib and the other drugs targeting YAP/TAZ pathway. Trametinib, in combination with VPA, resulted in synergistic activity in three of the cell lines tested (Figure 2A and Table 3). We found a synergistic activity of trametinib in combination with verteporfin for UPMM3, MEL270, and 92.1 cell lines (Figure 2B and Table 3).

### 3.3. The Combination of Trametinib and Cerivastatin Induces Apoptosis and Cell Cycle Arrest in UM Cell Lines

We selected the combination of trametinib with cerivastatin for further studies since cerivastatin showed the strongest effects among the statins and since HDACi and verteporfin have already been studied in depth. We decided to deepen the mechanisms of cerivastatin synergy with trametinib in the monosomic and *BAP1*-mutated UM cell lines (UPMM2 and UPMM3) representative of high-risk primary UM and on OMM2.5 derived from a liver metastasis (Table 1).

We studied the mechanisms of cell death induced by trametinib and cerivastatin by performing Annexin V/PI staining on UPMM2, UPMM3, and OMM2.5 cell lines. UPMM2 treated with an IC50 dose of either trametinib or cerivastatin showed an increase in the percentage of apoptotic cells of 30% and 20%, respectively, compared with controls. Combination of trametinib and cerivastatin at the synergistic concentration of 10.2 nM and 0.5 µM (half of trametinib IC50 and 6.6-fold less than cerivastatin IC50), respectively, induced 40% of apoptosis. A higher percentage of cells underwent apoptosis when the IC50s of the two drugs were combined (Figure 3A). The cell line UPMM3 showed a powerful increase of apoptosis when treated with trametinib and cerivastatin, comparable to that observed in cells treated with the combinations of the two drugs at their IC50. Finally, synergic doses of trametinib and cerivastatin induced apoptosis in OMM2.5 cells after seven days of incubation. Among the concentrations and combination tested, the IC50 of trametinib and cerivastatin induced the highest percentage of apoptosis (Figure 3A). The synergistic concentrations of trametinib in OMM2.5 were 32- and 108-times lower than its IC50 and the synergic doses of cerivastatin were 3.6- and 7.2-times lower than the IC50 of cerivastatin. The treatment at synergistic doses caused apoptosis comparable to the single IC50 of trametinib in OMM2.5 and UPMM2.

Treatment with trametinib blocked the cell cycle in the G0/G1 phase in all UM cells tested. Synergistic combinations of trametinib and cerivastatin caused the same cell cycle arrest observed when using the IC50 concentrations of trametinib (Figure 3B).

### 3.4. Trametinib and Cerivastatin Synergistic Concentrations Increase Apoptosis and Decrease Survival/Proliferation Pathways

We next focused on the possible signal transduction pathways involved in the increased cytotoxic activity mediated by the synergistic combination of trametinib and cerivastatin treatment. Cleaved PARP-1 and active caspase3 were only expressed in the synergistic combination (Figure 4A and Appendix A), suggesting a strong potentiation of cell death by the addition of cerivastatin to trametinib. More importantly, the expression of p-AKT was greatly reduced in the trametinib and cerivastatin treatment. As expected, phosphorylation of ERK1/2 kinases was downmodulated by treatment with trametinib and appeared to be further repressed by the combination of the two drugs. Finally, translocation of YAP into the nucleus, a requisite for its activity as a transcription factor, was not affected by trametinib but markedly reduced by cerivastatin as well as by the trametinib and cerivastatin treatment (Figure 4B and Appendix A).

### 3.5. Trametinib and Cerivastatin Synergistic Concentrations Down-Modulate Genes Involved in Cell Cycle Regulation and DNA Replication

We analysed the differential gene expressions of the UPMM3 cell line upon treatment with cerivastatin, trametinib and their synergistic combination. The heatmap of genes that were differentially expressed when comparing cells treated with the combined therapy versus cells treated with trametinib alone is shown in Figure 5A. The effect of cerivastatin on UPMM3 cells in combination with trametinib augmented the expression of genes related to the cholesterol synthesis pathway [47] (Figure 5A,B). Among the differentially expressed genes, *B4GALT5*, *IDH1* and *RASSF3* displayed an intermediate level of expression compared with cerivastatin and trametinib alone (Figure 5A). *B4GALT5* and *IDH1* are associated with tumour growth [48,49] and *RASSF3* is a tumour suppressor gene [50]. *SLC45A2*, a melanosomal transport protein coding gene [51] that is highly expressed in UM and present at very low levels in normal melanocytes [52] is down regulated in the combined therapy (Figure 5A). *RHOB* negatively regulates diverse cellular processes including cell cycle proliferation [47,53], and is strongly upregulated by the combined treatment (Figure 5A). The combination therapy also increased the expression of the *KRAS* gene signalling pathway in the cell line tested. Upregulation of *KRAS* likely reflects the low levels of post-translationally modified KRAS protein due to the inhibition of the mevalonate pathway by cerivastatin [47,53,54]. Yet, the strong reduction of p-ERK1/2 following the combined treatment (Figure 4A) indicates a lack of increased activity of the KRAS pathway. We studied the differential expression of genes of each treatment condition compared with control (Appendix A). In cerivastatin treated cells, we observed upregulation of genes involved in the steroid biosynthesis pathway, the activated AMP-Kinase (AMPK) and the peroxisome proliferator-activated receptor (PPAR) signalling pathways, consistent with anti-proliferative effects (Appendix A). Trametinib-treated cells showed upregulation of genes involved in AMPK and PI3K-AKT pathways and downmodulation of cell cycle, DNA repair and replication genes (Appendix A). The combined therapy induced upregulation of genes involved in the FoxO tumour suppressor and in the AMPK pathways, and down-modulation of genes involved in cell cycle regulation and DNA replication (Appendix A).

### 3.6. Trametinib and Cerivastatin Synergistic Concentrations Inhibit the Growth of the Monosomic BAP1 Mutated UM Cell Line In Vivo

In order to validate the effects of trametinib and cerivastatin combination on UM cells in vivo we generated xenografts of a primary monosomic *BAP1*-mutated human UM cell line, UPMM3, in highly immunodeficient *NOD.Cg-Prkdc^scid^ Il2rg^tm1wjl^/SzJ mice*. Chr3 monosomy and *BAP1* mutation are hallmarks of UM metastasis. For this reason, UPMM3 was considered the best model for this disease. One week after subcutaneous injection of UPMM3 cells, when the tumour was palpable, four *mice* per group were treated with vehicle, trametinib (1 mg/kg, *per os*, three days/week), cerivastatin (2 mg/kg *per os*, three days/week) or trametinib and cerivastatin for 57 days (until day 64). The end of treatment was followed by 15 days of observation. Strong inhibition of tumour growth was observed for trametinib-treated *mice*. The addition of cerivastatin determined a significantly stronger inhibition of tumour growth compared with *mice* treated with trametinib alone (*p* = 0.03, Figure 6A,B). *Mice* showed neither signs of toxicity nor loss of body weight (Figure 6C). During the observation period, the tumour resumed growth in trametinib-treated *mice*, while it remained not detectable in trametinib and cerivastatin-treated *mice*. Indeed, in days 71, 77 and 81, tumours from *mice* treated with trametinib and cerivastatin remained significantly smaller than those from trametinib group (*p* = 0.02, *p* = 0.04, *p* = 0.01, respectively, Figure 6A). Notably, *mice* treated with cerivastatin presented a trend to smaller tumours at the end of the treatment compared with control *mice* although this did not reach statistical significance.

## 4. Discussion

In stark contrast to what has been obtained for cutaneous melanoma, targeted therapy using MAP-kinase and MEK-inhibitors [11] and immunotherapy [56] have shown little to no activity in clinical trials for the treatment of metastatic uveal melanoma. This disappointing result is likely in part due to the different oncogenic signalling of the main UM driver mutations in *GNAQ* or *GNA11* that activate two different oncogenic signalling pathways, the MAP/MEK-kinase and the YAP/TAZ pathways. The latter, considered important for organ growth control, has only recently gained more attention in oncology. Oncogenic mutations of *GNAQ* and *GNA11* lead to dephosphorylation of YAP and TAZ, with nuclear translocation of YAP and increased cell growth and anti-apoptotic activity [57].

Given the activation of two different oncogenic pathways, the association of different classes of drugs targeting signalling downstream of GNAQ or GNA11 has been proposed to block UM cell growth [32,58]. Drugs targeting YAP/TAZ are natural candidates for combined treatments. Verteporfin, a photosensitizer used to treat the wet form of macular degeneration [59], has been shown to abolish YAP/TAZ signalling by disrupting the YAP–TEAD complex [44]. In uveal melanoma, the activity of verteporfin appears, however, to be more pronounced in cells carrying the wild-type form of *BAP1* than in cells in which BAP1 function is abolished by mutations and monosomy of chromosome 3, as is typical for UM with high metastatic risk [29]. In addition, verteporfin is not suited to systemic administration for prolonged times due to its toxic effects [60].

In our hands, verteporfin displayed synergistic activity with trametinib in three of the six cell lines tested, including the monosomic *BAP1*-mutated UPMM3 cell line. HDAC inhibitors such as VPA also have inhibitory effects on UM cell growth [33]. More recently, the HDAC4 inhibitor quisinostat was shown not only to reduce UM cell growth but also to induce expression of HLA class I, potentially positively influencing immunotherapy [61]. Quisinostat has also been investigated by Harbour et al. and found to prevent the growth of *BAP1*-mutant UM in a *mouse* model [62]. Though tested in translational studies for UM, quisinostat is, to the best of our knowledge, not being tested in clinical trials involving UM patients. Conversely, VPA is being tested in two clinical trials (NCT02068586, NCT04729543). NCT02068586 is testing the HDACi in the adjuvant setting enrolling 90 UM patients [63]. We show here that IC50 values for the HDAC inhibitor VPA are in the millimolar range for all six cell lines tested, in accordance with other reports [61,62]. In our hands, VPA showed synergy with trametinib in only three of the six cell lines tested (92.1, MEL270, and UPMM3).

Epidemiological studies have shown evidence of reduced cancer incidence and mortality with statins, though their anti-cancer properties remain under investigation [64]. High throughput screening of drug libraries identified the activity of cerivastatin on YAP to overcome the resistance of NSCLC cells to the ALK kinase inhibitor crizotinib [65]. A similar study identified cerivastatin as particularly potent in the induction of apoptosis of highly metastatic osteosarcoma cells [66]. We have explored, for the first time, the possibility of combining MEK inhibitors with statins, which interfere with YAP oncogenic activity by inhibiting the mevalonate pathway, thereby affecting geranyl-geranylation and subcellular localization of Rho-GTPase [27].

We have shown that statins have antiproliferative effects in the majority of the UM cell lines tested. The most potent effect was obtained with cerivastatin. We observed synergistic effects of trametinib and cerivastatin in UPMM3 and UPMM2 monosomic and *BAP1*-mutated UM cell lines, models of UM at high risk of metastasis. OMM2.5 cells, derived from an UM liver metastasis, are also sensitive to the combination of trametinib and cerivastatin, although at a lower level compared with the monosomic/*BAP1* mutated cells. The simultaneous exposure of UM cells to cerivastatin and the MEK inhibitor allowed us to obtain the IC50 at trametinib doses 100-times lower in the combination, than with trametinib alone. This translates into enhanced efficacy with possibly more manageable toxicity when performing clinical trials. The synergic concentration of cerivastatin was six-fold lower than the IC50 value of the drug alone, an important fact since cerivastatin has been discontinued for clinical use as a cholesterol-lowering medication due to its toxic effects [67].

Transplantation of the UPMM3 cell line in highly immunologically deficient *mice* led to the development of tumours in 100% of *mice*. The oral administration of trametinib and cerivastatin significantly reduced tumour growth compared with control and to the two drugs used as a single treatment with no evidence of toxicity. Cerivastatin alone did not significantly affect tumour growth. Trametinib alone, instead, reduced the size of tumours in *mice*, confirming an inhibitory effect of trametinib on UM growth in vivo [68]. However, 15 days after the end of treatment, tumour growth was still inhibited in *mice* treated with trametinib and cerivastatin but not in the *mice* treated with trametinib used as a single drug.

The synergistic treatment in vitro of UPMM3 cells affected YAP translocation into the nucleus and involved inhibition of ERK1/2 activation and AKT signalling. Notably, the upregulation of AKT signalling, a potential mechanism of escape from MEK inhibition [45], is blocked by adding cerivastatin to trametinib. The ability to block the cell cycle in the G0/G1 phase, an increase in apoptosis, and the expression of the cleaved form of Poly(ADP-Ribose) Polymerase 1 (PARP1) and active caspase3 (CASP3) characterize the synergistic drug combination compared with the single drug treatments. Micro-array analysis of UPMM3 cells treated with a synergistic combination of trametinib and cerivastatin showed a modification of gene expression compared with trametinib alone. A strong up-regulation of genes involved in lipid metabolism is possibly due to cerivastatin blocking the mevalonate pathway as reported by others [47,53].

The reduced expression of *RASSF3*, a paralog of *RASSF1* that has already been implicated as a tumour suppressor in UM [69], following treatment with trametinib is partially recovered by the addition of cerivastatin. RASSF1 and 3 contain a SARAH-domain that intervenes in the regulation of MST1 and MST2 kinases in the activation of YAP [70,71], an activity crucial to its tumour suppressor function [72]. *KRAS* and *RAS*-related genes appear upregulated probably as a consequence of the unavailability of the correct post-translationally modified proteins caused by the mevalonate block [47,53]. All statins tested showed synergistic activity in combination with trametinib, yet cerivastatin yielded the most convincing effects. Cerivastatin has been commercialized as Lipobay, a cholesterol-lowering agent for the prevention of cardiovascular disease, as with other statins. It was withdrawn following reports of 52 cases of fatal rhabdomyolysis that occurred ten times more frequently than with other statins [73]. Rhabdomyolysis is a toxic effect common to all statins but 10 to 50 times stronger for cerivastatin [74]. In many cases, fatal outcomes occurred in patients concomitantly treated with other lipid-lowering medications such as lovastatin [74] or gemfibrozil [73]. The exact mechanism of induction of rhabdomyolysis is still unknown. However, it is likely to depend on the effects of statins on the mevalonate pathway [75]. *Mice* in which the gene encoding the molecular target of statins, HMG-CoA reductase, has been disrupted, develop rhabdomyolysis spontaneously [76]. Cerivastatin shows five to 250 times stronger lipid-lowering activity than other statins [77]. Therefore, the strong cerivastatin-associated rhabdomyolysis most likely depends on potent effects on the mevalonate pathway that, as we show here, also determine stronger activity in blocking UM proliferation.

When considering cerivastatin for treating MUM, the patient’s specific risk profile must be considered. Patients at high risk of developing statin-induced rhabdomyolysis can be identified [78,79]. The rhabdomyolysis risk must be weighed against the risk of succumbing to uveal melanoma metastases and the potential benefit of the therapy. Using a statin associated with a lower risk of rhabdomyolysis will likely not help since it will also be less active against metastatic uveal melanoma.

## 5. Conclusions

We propose to test the combination of trametinib and cerivastatin in clinical trials. This combination could improve the survival of metastatic uveal melanoma patients while waiting for the development of specific GNAQ and GNA11 inhibitors and in the rarer cases of GNAQ/11 wild-type metastatic uveal melanoma.

## Figures and Tables

**Figure 1 cancers-15-00886-f001:**
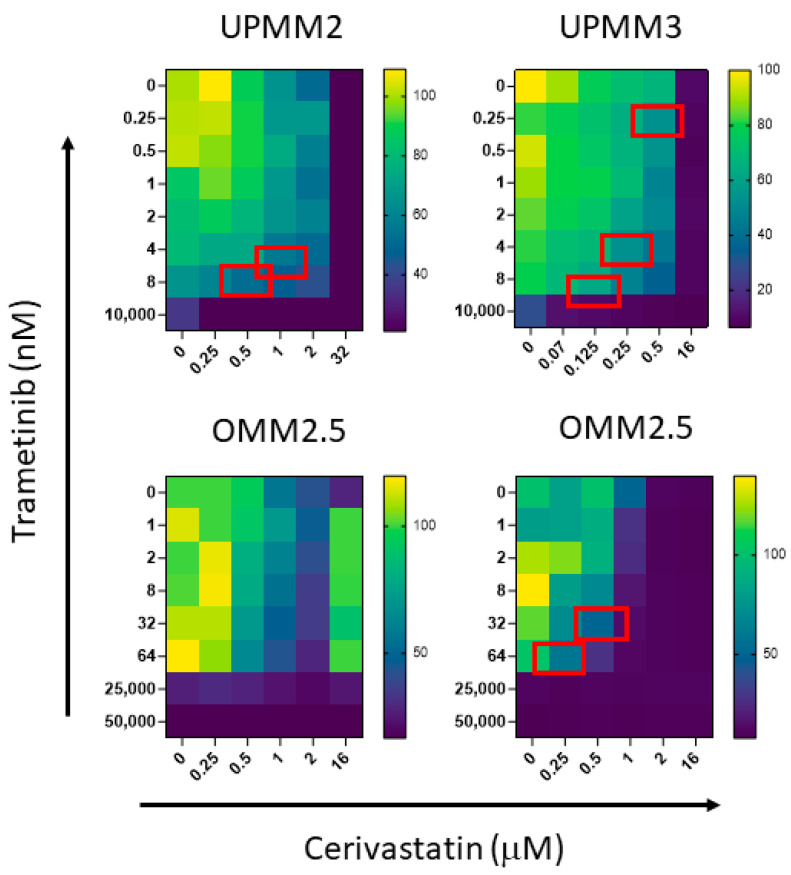
Minimal dose–response matrices showing trametinib-based combinations with cerivastatin by MTT assay. Synergistic combinations as calculated using the Chou–Talalay method are indicated by red boxes. Cell viability is reported by a colour code ranging from yellow (100%) to dark blue (<20%) as indicated by the bar on the right. OMM2.5 were tested at 72 h (lower left) and one week (lower right) of treatment.

**Figure 2 cancers-15-00886-f002:**
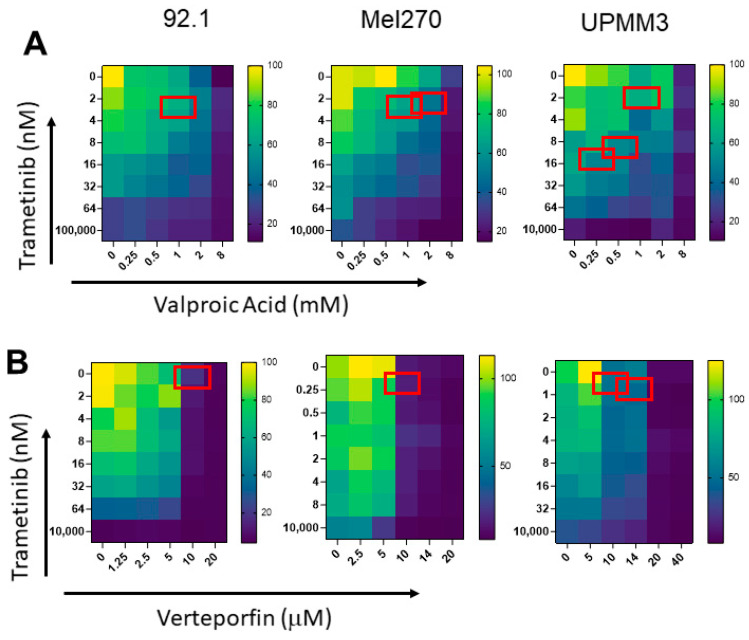
Minimal dose–response matrices showing trametinib-based combinations with VPA (**A**) and verteporfin (**B**) by MTT assay, at 72 h. Synergistic combinations, as calculated using the Chou–Talalay method, are indicated by red boxes. Cell viability is reported by a colour code ranging from yellow (100%) to dark blue (<20%) as indicated by the bar on the right.

**Figure 3 cancers-15-00886-f003:**
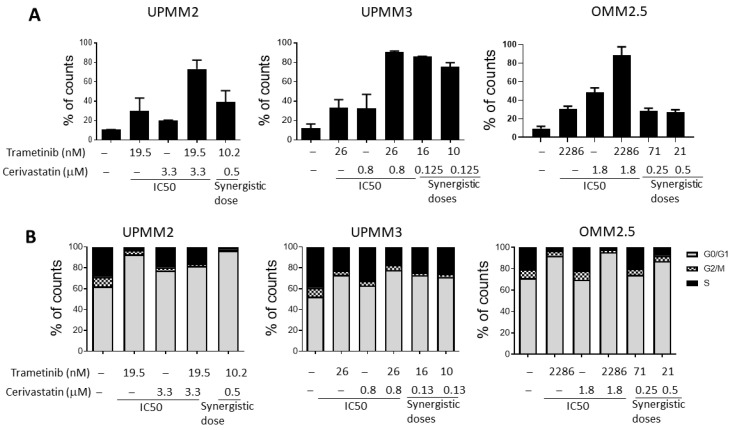
Percentage of apoptotic UM cells (Annexin V + PI+ and Annexin V + PI−) after treatment with IC50 or the synergistic concentrations of trametinib and cerivastatin (**A**). Data are expressed as mean percentage ± SEM. Stacked histograms showing UM cells distribution in cell cycle phases after treatment for 72 h with IC50 and synergistic concentrations of trametinib and cerivastatin. Data from one representative experiment out of three are reported (**B**).

**Figure 4 cancers-15-00886-f004:**
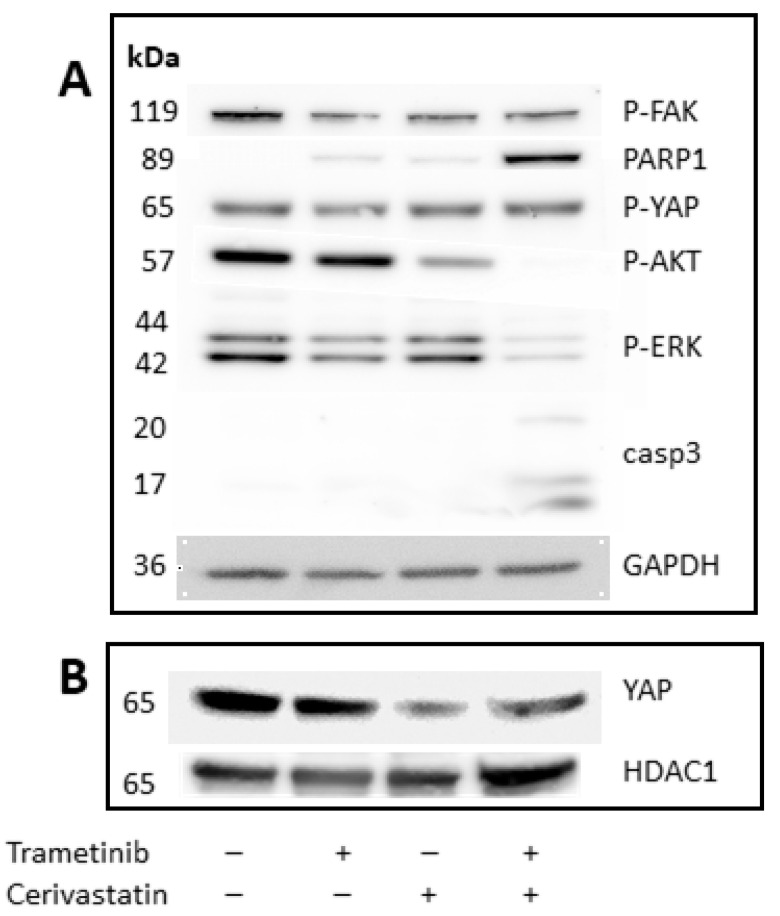
Western blot analysis of cytoplasmic (**A**) and nuclear (**B**) protein extracts of UPMM3 treated with no drug, with trametinib at the synergistic dose of 10 nM, cerivastatin at the synergistic dose of 0.125 µM and with the combination of trametinib and cerivastatin, for 24 h. GAPDH and HDAC1 are the loading controls for cytoplasmic and nuclear extracts, respectively.

**Figure 5 cancers-15-00886-f005:**
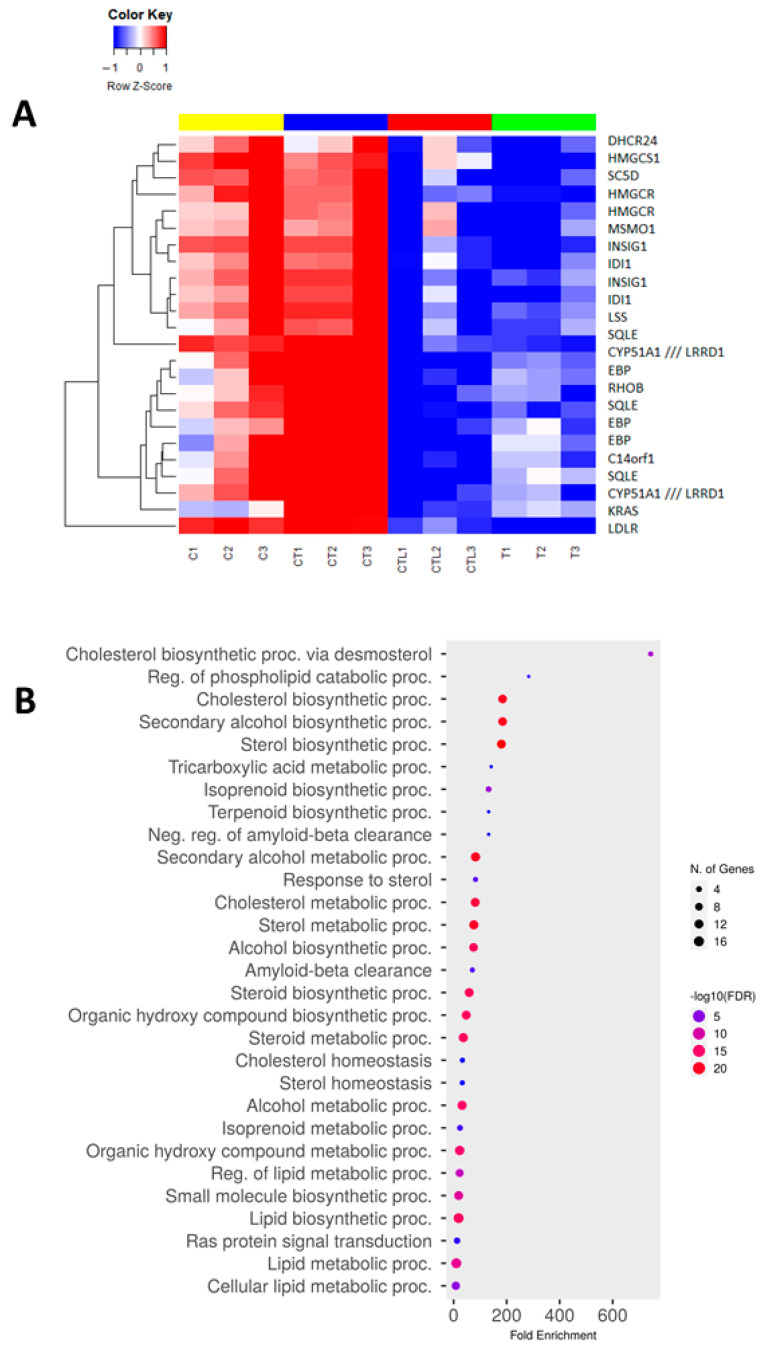
Expression profiles of UPMM3 cells treated with the combination of trametinib (10 nM) and cerivastatin (0.125 µM) versus trametinib, for 24 h. (**A**): The expression profiles were interrogated by significance analysis of micro-arrays and the expression values of significantly differentially expressed genes were clustered by hierarchical clustering. The expression values are reported by a colour scale (blue = expression below the mean, red = expression above the mean, white = expression at the mean; the intensity is related to the distance from the mean). The bars above the dendrogram show the treatment status (cerivastatin = yellow, untreated = red, trametinib = green, and trametinib and cerivastatin treatment = blue). (**B**): Gene set enrichment analysis for statistically significant GO biological process related to the above identified differentially expressed genes. The X-axis label represents fold enrichment = amount of differentially expressed genes enriched in the GO/amount of all genes in the background gene set and the Y-axis label lists Gene Ontology Biological Process categories (GO_BP). The size and colour of the bubble represent the amount of differentially expressed genes enriched in the GO_BP and the enrichment significance (false discovery rate calculated based on nominal *p*-value from the hypergeometric test), respectively. The closer the colour is to red, the more significant is the enrichment. The highest confidence levels are shown (ShinyGO [55] 0.76.2 http://bioinformatics.sdstate.edu/go/ accessed on 26 January 2023).

**Figure 6 cancers-15-00886-f006:**
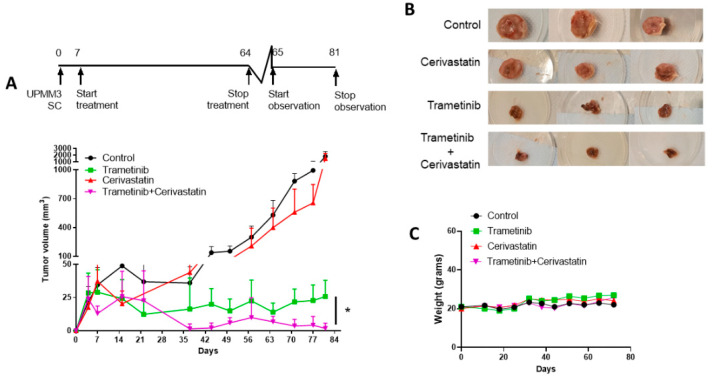
In vivo tumour growth of UPMM3 injected subcutaneously in *NOD.Cg-Prkdc^scid^ Il2rg^tm1wjl^/SzJ mice*. *Mice* treated with vehicle alone (control, black circle), trametinib (green square), cerivastatin (red up-pointing triangle) and their combination (magenta down-pointing triangle). The X-axis shows days of measurements and the Y-axis tumour volume in mm^3^. A schematic representation of the experimental design is shown above (**A**). Representative images of subcutaneous tumours from *NOD.Cg-Prkdc^scid^ Il2rg^tm1wjl^/SzJ mice* for each group of treatment (**B**). Weight changes in grams between different groups of treatment (**C**). * *p* = 0.03.

**Table 1 cancers-15-00886-t001:** Uveal melanoma cell line characteristics.

Name	Origin	GNAQ	GNA11	BAP1	chr3 Status	Reference
92.1	primary	Q209L	WT	WT	trisomic	[34]
MEL270	primary ^$^	Q209P	WT	WT	disomic	[35]
OMM1	metastatic	Q209L	Q209L	WT	disomic	[36]
OMM2.5	metastatic ^$^	Q209P	WT	WT	disomic	[37]
UPMM2	primary	Q209L	WT	I586H fs * 57	monosomic	[38]
UPMM3	primary	Q209P	WT	G45F48del	monosomic	[38]

$ = from the same patient; fs * 57 = frame shift with 57 unrelated amino acids; WT = wild type.

**Table 2 cancers-15-00886-t002:** IC50 values of the drugs tested in UM cell lines.

			OMM2.5		UPMM3		MEL270		92.1		OMM1		UPMM2	
Drug	Class	Unit	IC50	SEM	IC50	SEM	IC50	SEM	IC50	SEM	IC50	SEM	IC50	SEM
trametinib	MEK-inhibitor	nM	171.60	±44	16.39	±6	5.00	±1.4	9.90	±4.8	6500.00	±3983	8.10	±3
valproic acid	HDAC-inhibitor	mM	40.90	±26.5	2.77	±93	10.88	±2.56	3.39	±0.98	9.56	±2.73	23.24	±2.34
verteporfin	Photosensitizer	µM	8.22	±3.88	30.24	±1.44	18.67	±0.9	16.55	±0.788	13.15	±4.75	720.60	±703.5
cerivastatin	Statin	µM	0.99	±0.16	0.35	±0.1	0.40	±0.09	1.56	±0.26	0.06	±0.02	0.38	±0.12
atorvastatin	Statin	µM	62.11	±46.45	116.50	±96.26	137.30	±55.17	na	na	5.63	±0.62	50.75	±20.84
simvastatin	Statin	µM	8.29	±0.8	8.32	±1.12	15.57	±5.9	115.40	±32.6	2.83	±0.34	36.17	±14.6

**Table 3 cancers-15-00886-t003:** Synergisms of trametinib with VPA (A), verteporfin (B) and cerivastatin (C) are shown. UM cell lines were treated for 72 h or seven days (*). The IC50 concentrations are reported as well as the minimum concentrations at which synergism was observed (synergistic concentration). The combination index was calculated according to Chou–Talalay (see text). A combination index < 1 indicates synergism. Only data for cell lines for which synergism could be observed are shown.

**A: Trametinib and Valproic Acid (VPA)**					
**Cell Line**	**Trametinib**	**VPA**	**Trametinib**	**VPA**	**Combination Index**
	**IC50**		**Synergistic Concentration**		
	**nM**	**mM**	**nM**	**mM**	
92.1	14.60	5.56	3.64	1.00	0.43
Mel270	7.13	14.65	2.24	1.00	0.38
Mel270	7.13	14.65	1.77	2.00	0.38
UPMM3	20.38	1.84	13.02	0.25	0.77
UPMM3	20.38	1.84	9.61	0.50	0.74
UPMM3	20.38	1.84	2.39	1.00	0.65
**B: Trametinib and Verteporfin**					
**Cell Line**	**Trametinib**	**Verteporfin**	**Trametinib**	**Verteporfin**	**Combination Index**
	**IC50**		**Synergistic Concentration**		
	**nM**	**µM**	**nM**	**µM**	
92.1	26.74	15.76	0.15	10.00	0.63
Mel270	1.20	17.78	0.00	10.00	0.56
UPMM3	11.08	28.84	0.43	10.00	0.39
UPMM3	11.08	28.84	0.55	14.00	0.53
**C: Trametinib and Cerivastatin**					
**Cell Line**	**Trametinib**	**Cerivastatin**	**Trametinib**	**Cerivastatin**	**Combination Index**
	**IC50**		**Synergistic Concentration**		
	**nM**	**µM**	**nM**	**µM**	
UPMM2	19.53	2.66	10.20	0.50	0.70
UPMM2	19.53	2.66	6.15	1.00	0.68
UPMM3	26.31	0.77	6.00	0.25	0.54
UPMM3	26.31	0.77	0.27	0.50	0.66
OMM2.5 *	2286.00	1.11	71.00	0.25	0.25
OMM2.5 *	2286.00	1.11	21.00	0.50	0.44

* Cells tested after one week of treatment.

## Data Availability

All micro-array data are MIAME compliant. The dataset, corresponding to 12 data files is available from the GEO database (http://www.ncbi.nlm.nih.gov/geo/, accessed on 26 January 2023), under accession number GSE212219.

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
