# Peer review of "Cerivastatin Synergizes with Trametinib and Enhances Its Efficacy in the Therapy of Uveal Melanoma"

_cancers, 2023, doi:10.3390/cancers15030886_

Round 1

Reviewer 1 Report

The study of Amaro et al investigates synergistic effects of drugs on uveal melanoma cells. They identify Trametinib and Cerivastatin as a potential synergistic drug combination with improved outcome over each substance alone. The study is of high interest given that no good treatment option exists for UM.

1) Some of the figures are low quality, with small font sizes, also some white patches in figure 6a. The quality and readability of the figures should be improved.

2) The red circles in figure 2 are not ideal, as they overlap with parts of multiple concentrations. Maybe squares or another mean of marking synergistic concentrations is better.

3) Error bars for the mouse data seem to be missing in Figure 6a

4) Which days are significant for the combo treatment in Figure 6a? The significance label does not clarify that.

5) What HDAC is stained for on the western blot?

6) Given that Quisinostat shows much lower IC50 values what was the reason to use VPA instead?

Author Response

Point by point response to the reviewer:

Reviewer #1:

The study of Amaro et al investigates synergistic effects of drugs on uveal melanoma cells. They identify Trametinib and Cerivastatin as a potential synergistic drug combination with improved outcome over each substance alone. The study is of high interest given that no good treatment option exists for UM.

  • Some of the figures are low quality, with small font sizes, also some white patches in figure 6a. The quality and readability of the figures should be improved.

We increased the quality, font sizes and in some case also the size of the figures.

  • The red circles in figure 2 are not ideal, as they overlap with parts of multiple concentrations. Maybe squares or another mean of marking synergistic concentrations is better.

We substituted the circles by squares in the figures as suggested, now Figures 1 and 2

  • Error bars for the mouse data seem to be missing in Figure 6a

We added the error bars that were omitted by error.

  • Which days are significant for the combo treatment in Figure 6a? The significance label does not clarify that.

We found statistically significative the difference between trametinib and combo treatment at the Day 44, 71, 77, and 81. We added this information in the text in the description of figure 6a (lines 502-504). The tumor growth curve was significantly different in the two groups (p=0.03 Figure 6A, lines 496-498).

  • What HDAC is stained for on the western blot?

Anti-HDAC1-antibody was used. We added this information to the figure, legend to figure 5 and in the methods section.

  • Given that Quisinostat shows much lower IC50 values what was the reason to use VPA instead?

Valproic acid is being tested in two clinical trials (NCT02068586, NCT04729543). NCT02068586 is testing the HDAC-inhibitor in the adjuvant setting enrolling 90 UM patients. Quisinostat, though tested in translational studies for UM, is to the best of our knowledge not being tested in clinical trials involving UM patients. We added a comment in the discussion section.

Reviewer 2 Report

The manuscript by Amaro et al., "Cerivastatin Synergizes with Trametinib and Enhances its 2 Efficacy in the Therapy of Uveal Melanoma" is an interesting study showing that the association of cerivastatin with trametinib may improve its efficacy in the treatment of uveal melanoma.

The introduction presents a good presentation of the mechanisms known to be involved in uveal melanoma. Unfortunately, the work is presented as being merely descriptive of the effects of a set of drugs that supposedly act on the YAP/TAZ pathway, in response to trametinib. On this point, the authors should present, in a more assertive way, the objective of the work.

The Materials and Methods section is well presented, with only two points that should deserve the authors' attention. (i) The use of MTT for the determination of cell proliferation is questionable since what is being measured is the metabolic activity. This point deserved to be discussed in the interpretation of the results.

(ii) The method described by Chou-Talalay is used for the study of drug combinations. Not being a common method, and so important for the present work, it should be better described so the reader can better interpret the results.

The first paragraph of the Results section should be referred to the discussion supported by the references that support the effect of the drugs in the YAP/TAZ pathway.

Interestingly, the problem to be addressed supposedly in the study seems to be here clearly indicated: as "MAPK targeted therapy has so far not been successful in UM as a single agent" (line 211), the objective of the work would be to find drugs that can be combined with inhibitors of MAPK to improve its effectiveness. Is it correct?

The authors show that the sensitivity of different cell lines to trametinib is variable, with OMM, especially OMM1, being the least sensitive. 

Why the OMM1 line is not used in interaction studies afterwards? Although the EC50 value is higher (as shown in Table 2), this value (6.5 µM) is still lower than that of valproic acid, verteporfin and atorvastatin. It would make sense to the reader that this would be the preferred cell line for studying interactions. The reasons why it was not should be better explained.

The use of other methods to demonstrate the value of the association with trametinib strengthens the work. Unfortunately, the authors do not use a coherent strategy in the selection of drugs and cell lines. If the objective was to find associations that potentiate the effects of trametinib in less sensitive cells, why not using less sensitive cell lines and started using the UPMM lines? 

Other questions arise that the authors should explain: why were other drugs abandoned and interaction studies continued only with cerivastatin? Although it was the drug that obtained the lowest Combination index in OMM2.5 cells, this drug was withdrawn from the market, unlike valproic acid, verteporfin and other statins. Has this option anything to do with the more serious rhabdomyolysis it caused, when compared with other statins?

The focus that is given from point 3.4 onwards creates a profound rupture with the rest of the work and a dissonance with the introduction that removes all coherence from the work.

The results obtained in the mechanisms of death and cell cycle of the association of trametinib with cerivastatin, as well as the assay in xenografts are well elaborated and the results are consistent. But they lack the connection with the first part.

Thus, in my opinion, the work presents is very interesting set of results but has a major flaw that consists of a lack of alignment described above with the objective and with the first part of the work.

If the aim is to study the interaction between cerivastatin and trametinib in uveal melanoma cell lines and deepen the mechanism of this interaction, the inclusion of other drugs does not add anything. If this option is chosen, authors will be required to make an extra effort to explain why they chose the UPMM lines for the studies described in 3.4, 3.5 and 3.6. and the discussion be aligned according to this rationale.

Other points:

Figure 1 is difficult to represent, and the data are already shown in Table 2. It must be removed.

The caption for Table 3 is not clear. The authors refer to the text to better understand how the synergistic concentration was calculated. However, in the text the explanation is very limited (We used the method described by Chou and Talalay [39] to calculate the combination index (CI) for additive effect (CI = 1), synergism (CI < 1), and antagonism ( CI > 1); lines 257-9). Is this the explanation you are referring to? If so, it does not help the reader to follow the results.

Figure 5 is too small. It is illegible.

The legibility problem is common to all figures. Authors should increase the font size.

A minor point to be aware of is NOT to use capital letters for drug names.

Author Response

Point by point to Reviewer #2

The manuscript by Amaro et al., "Cerivastatin Synergizes with Trametinib and Enhances its 2 Efficacy in the Therapy of Uveal Melanoma" is an interesting study showing that the association of cerivastatin with trametinib may improve its efficacy in the treatment of uveal melanoma.

-The introduction presents a good presentation of the mechanisms known to be involved in uveal melanoma. Unfortunately, the work is presented as being merely descriptive of the effects of a set of drugs that supposedly act on the YAP/TAZ pathway, in response to trametinib. On this point, the authors should present, in a more assertive way, the objective of the work.

We rephrased the last paragraph of the introduction in order to better outline the objective of the work.

-The Materials and Methods section is well presented, with only two points that should deserve the authors' attention. (i) The use of MTT for the determination of cell proliferation is questionable since what is being measured is the metabolic activity. This point deserved to be discussed in the interpretation of the results.

We added the following sentence to the results section: “The MTT-assay was used to measure cellular metabolic activity as an indi-cator of cell viability, proliferation and cytotoxicity. Since none of the drugs is expected to directly affect mitochondrial activity the formation of formazan can be used as a proxy for proliferation in UM cell lines.” (lines 254-258)

-(ii) The method described by Chou-Talalay is used for the study of drug combinations. Not being a common method, and so important for the present work, it should be better described so the reader can better interpret the results.

We added a more exhaustive explanation of the method to the methods section (lines 158-162).

-The first paragraph of the Results section should be referred to the discussion supported by the references that support the effect of the drugs in the YAP/TAZ pathway.

We added a reference to the discussion section in the first paragraph of the results section and added relevant references to the statements.

-Interestingly, the problem to be addressed supposedly in the study seems to be here clearly indicated: as "MAPK targeted therapy has so far not been successful in UM as a single agent" (line 211), the objective of the work would be to find drugs that can be combined with inhibitors of MAPK to improve its effectiveness. Is it correct?

This is correct. We added this statement to the sentence indicated.

-The authors show that the sensitivity of different cell lines to trametinib is variable, with OMM, especially OMM1, being the least sensitive. Why the OMM1 line is not used in interaction studies afterwards? Although the EC50 value is higher (as shown in Table 2), this value (6.5 µM) is still lower than that of valproic acid, verteporfin and atorvastatin. It would make sense to the reader that this would be the preferred cell line for studying interactions. The reasons why it was not should be better explained.

This is an important point. OMM1 cell lines were developed from non-hepatic metastases of UM while hepatic metastases are most frequent for UM and, importantly, usually determine therapy failure. Furthermore, metastatic UM is characterized by monosomy of chr3 and BAP1 mutation, two conditions not met by OMM1 cell line. Therefore, we focused our experiments on monosomic and BAP1 mutated UPMM3 cell line. Since in vivo testing of multiple cell lines is not considered ethical, we opted for testing the most relevant UM model. We added a comment in the results section (lines 348-351).

-The use of other methods to demonstrate the value of the association with trametinib strengthens the work. Unfortunately, the authors do not use a coherent strategy in the selection of drugs and cell lines. If the objective was to find associations that potentiate the effects of trametinib in less sensitive cells, why not using less sensitive cell lines and started using the UPMM lines?

As stated above, we chose UPMM lines since chr3 monosomy and BAP-1 mutations are hallmarks of metastatic UM.

-Other questions arise that the authors should explain: why were other drugs abandoned and interaction studies continued only with cerivastatin? Although it was the drug that obtained the lowest Combination index in OMM2.5 cells, this drug was withdrawn from the market, unlike valproic acid, verteporfin and other statins. Has this option anything to do with the more serious rhabdomyolysis it caused, when compared with other statins?

HDACi and verteporfin have been proposed by several translational studies for the treatment of metastatic UM and the HDACi valproic acid is being tested in a clinical trial in the adjuvant setting. To the best of our knowledge, there are no translational studies on statins and their combinations with MAPKi. Hence, we focused on the latter in the study in vivo. Analyses in vitro were performed on verteporfin and valproic acid for comparison. We think this part of the work contributes to put the effect of statins in the correct context but we would accept to move it to the supplementary information published with the article. The fact that cerivastatin has been withdrawn from the market, as explained in the discussion section, probably relies on its relatively strong effect. Rhabdomyolysis, cholesterol lowering activity and inhibition of geranyl-geranylation of Rho GTPases all rely on the same activity so that strong Rho GTPase inhibitors also have a higher risk of rhabdomyolysis. However, fatal rhabdomyolysis in some patient among millions treated is an unacceptable risk for a preventive drug but the risk must be evaluated considering the potential benefit for patients of metastatic UM. Most cancer therapies show a much higher toxicity and rate of fatalities.

-The focus that is given from point 3.4 onwards creates a profound rupture with the rest of the work and a dissonance with the introduction that removes all coherence from the work.

We rebuilt point 3.2 highlighting the importance of cerivastatin and trametinib combination and compared to the other combinations (lines: 286-296), consequently we switched figure 1 and 2. We added a comment at the beginning of point 3.3 to explain the selection of the drugs to be studied further (lines 343-345).

-The results obtained in the mechanisms of death and cell cycle of the association of trametinib with cerivastatin, as well as the assay in xenografts are well elaborated and the results are consistent. But they lack the connection with the first part.

See above

-Thus, in my opinion, the work presents is very interesting set of results but has a major flaw that consists of a lack of alignment described above with the objective and with the first part of the work. If the aim is to study the interaction between cerivastatin and trametinib in uveal melanoma cell lines and deepen the mechanism of this interaction, the inclusion of other drugs does not add anything. If this option is chosen, authors will be required to make an extra effort to explain why they chose the UPMM lines for the studies described in 3.4, 3.5 and 3.6. and the discussion be aligned according to this rationale.

As stated above, we think that the study of HDACi and verteporfin is pertinent but we are ready to move it to the supplementary data section, if required. Moreover, we stated in the results why we chose UPMM cells for mechanistic and in vivo experiments.

Other points:

-Figure 1 is difficult to represent, and the data are already shown in Table 2. It must be removed.

We agree with the Reviewer and removed figure 1. 

-The caption for Table 3 is not clear. The authors refer to the text to better understand how the synergistic concentration was calculated. However, in the text the explanation is very limited (We used the method described by Chou and Talalay [39] to calculate the combination index (CI) for additive effect (CI = 1), synergism (CI < 1), and antagonism ( CI > 1); lines 257-9). Is this the explanation you are referring to? If so, it does not help the reader to follow the results.

We added a more extensive description of the method to the methods section (lines 158-162).

-Figure 5 is too small. It is illegible.

We improved the legibility of figure 5.

-The legibility problem is common to all figures. Authors should increase the font size.

We improved legibility and font size where required.

-A minor point to be aware of is NOT to use capital letters for drug names.

We corrected the drug names.

Round 2

Reviewer 2 Report

The alterations improved significantly the manuscript. The message is now better understood and made the study more coherent. 

At this stage, I recommend only minor changes:

1) In my view, the comments on expected effects of drugs on MTT assay (lines 243 - 247) should be placed in the methods section (line 147).

2) Legends of figures 1 and 2 should be altered since now circles are no longer used.

3) The text should be review more carefully to correct some writing inconsistencies. Names of some drugs still appear written with capital letters. 

Author Response

Point by point response to the reviewer:

The alterations improved significantly the manuscript. The message is now better understood and made the study more coherent.

At this stage, I recommend only minor changes:

  • In my view, the comments on expected effects of drugs on MTT assay (lines 243 - 247) should be placed in the methods section (line 147).

Re: we moved the sentences “The MTT-assay was used to measure cellular metabolic activity as an indicator of cell viability, proliferation and cytotoxicity. Since none of the drugs is expected to directly affect mitochondrial activity, the formation of formazan can be used as a proxy for proliferation in UM cell lines.” in the methods section (2.1) as requested

  • Legends of figures 1 and 2 should be altered since now circles are no longer used.

Re: we apologize and corrected the legends

  • The text should be review more carefully to correct some writing inconsistencies. Names of some drugs still appear written with capital letters.

Re: we corrected the name of drugs through the text